# Ess-InfoGAIL: Semi-supervised Imitation Learning from Imbalanced Demonstrations

**Huiqiao Fu**[1], **Kaiqiang Tang**[1], **Yuanyang Lu**[1], **Yiming Qi**[1], **Guizhou Deng**[1],
**Flood Sung**[2], **Chunlin Chen**[1]*

[1]Nanjing University, China, [2]Moonshot AI, China

{hqfu, kqtang, yylu, ymqi}@smail.nju.edu.cn,
{guizhoudeng, floodsung}@gmail.com, clchen@nju.edu.cn

## Abstract

Imitation learning aims to reproduce expert behaviors without relying on an explicit reward signal. However, real-world demonstrations often present challenges, such as multi-modal, data imbalance, and expensive labeling processes. In this work, we propose a novel semi-supervised imitation learning architecture that learns disentangled behavior representations from imbalanced demonstrations using limited labeled data. Specifically, our method consists of three key components. First, we adapt the concept of semi-supervised generative adversarial networks to the imitation learning context. Second, we employ a learnable latent distribution to align the generated and expert data distributions. Finally, we utilize a regularized information maximization approach in conjunction with an approximate label prior to further improve the semi-supervised learning performance. Experimental results demonstrate the efficiency of our method in learning multi-modal behaviors from imbalanced demonstrations compared to baseline methods.

## 1 Introduction

The recent success of Reinforcement Learning (RL) has shown great potential in solving complex decision-making tasks [1, 2, 3]. A key prerequisite for RL is to design a reward function, which can be challenging in many real-world applications [4, 5]. By contrast, Imitation Learning (IL) methods, such as Behavior Cloning (BC) [6], Inverse Reinforcement Learning (IRL) [7] and Generative Adversarial Imitation Learning (GAIL) [8], can reproduce expert behaviors without relying on an explicit reward signal. Among them, GAIL performs a more effective learning process by learning a policy directly from demonstrations, and has been widely applied in various fields [9, 10, 11, 12].

Real-world demonstrations are typically coupled by multi-modal behaviors. To disentangle interpretable and meaningful behavior representations, InfoGAIL [13] utilizes the concept of maximizing mutual information between discrete or continuous latent variables to generate corresponding samples in an unsupervised manner. However, unsupervised learning of desired decodable factors from unstructured data can be challenging, particularly in the case of behavior imitation, where commonly used methods such as contrastive learning often underperform. Recent researches have uncovered that unsupervised learning of disentangled representations is inherently difficult without the presence of inductive biases on both the models and data [14]. Additionally, real-world demonstrations often consist of imbalanced behavior modes. For example, walking constitutes the majority of human behavior, whereas jumping only represents a small fraction. It is difficult for InfoGAIL to learn disentangled behavior representations from imbalanced demonstrations, as shown in Fig. 1. One intuitive solution is to introduce supervised learning with fully labeled data [15]. However, manually segmenting and labeling from the raw demonstrations can be a costly endeavor.

---

*Corresponding author

37th Conference on Neural Information Processing Systems (NeurIPS 2023).

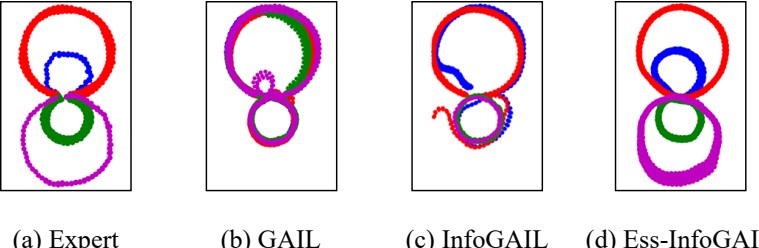

| (a) Expert | (b) GAIL | (c) InfoGAIL | (d) Ess-InfoGAIL |

Figure 1: 2D trajectory environment. An agent can move freely within a plane, seeking to mimic 4 expert trajectories, each represented by a different color to indicate a distinct mode of expert behavior. (a) The imbalanced expert demonstrations, where the red and green trajectories are predominant, while the blue and purple trajectories only represent a small portion. (b) Trajectories generated by GAIL. (c) Trajectories generated by InfoGAIL. (d) Trajectories generated by the proposed Ess-InfoGAIL. Our method is able to learn high quality disentangled behavior modes from the imbalanced expert demonstrations.

To tackle these challenges, we introduce a novel semi-supervised imitation learning architecture that can acquire disentangled behavior representations from imbalanced demonstrations, guided by only a limited amount of labeled data (as little as 0.5% to a maximum of 2% of the dataset). Our model, Elastic semi-supervised InfoGAIL (Ess-InfoGAIL), makes three key improvements to the related InfoGAIL. First, we draw inspiration from ss-InfoGAN [16] and focus on learning meaningful and controllable behavior representations by maximizing two mutual information terms: i) The mutual information between a latent skill variable, drawn from a categorical distribution, and the labeled expert state-action transitions, ii) and the mutual information between a latent shifting variable, drawn from a continuous uniform distribution, and the state-action transitions produced by a policy. The former term enforces that the latent skill variable corresponds to label categories in a semi-supervised manner, while the latter allows for style shifting within a given skill. Second, we tackle the issue of imbalanced data and align the distribution of generated transitions with that of expert demonstrations by utilizing a latent skill variable drawn from a differentiable Gumbel-Softmax distribution [17], instead of using a latent skill variable drawn from a fixed categorical distribution. Third, to make use of the intrinsic information in the unlabeled imbalanced data and improve the efficiency of the semi-supervised learning, we leverage the Regularized Information Maximization (RIM) [18] with a label prior that approximates the learned latent skill distribution.

In the experiment, we first validated our method in a simple 2D trajectory environment to obtain a more intuitive visualization, as shown in Fig. 1, and subsequently tested our method in 4 challenging MuJoCo environments (Reacher, Pusher, Walker-2D, and Humanoid). The experimental results illustrate the efficiency of our method in disentangling multi-modal behaviors from imbalanced demonstrations compared to baseline methods. To the best of our knowledge, our work is the first to tackle the problem of learning disentangled behavior representations from imbalanced demonstrations, with the aid of only a limited amount of labeled data. The code is available at `https://github.com/tRNAoO/Ess-InfoGAIL`. In particular, the contributions of this work are threefold:

1. We extend the InfoGAIL with a semi-supervised imitation learning architecture, resulting in Ess-InfoGAIL, which enables the learning of desired disentangled behavior representations.

2. We utilize a learnable latent distribution to align the generated and expert distributions in the imbalanced data setting, and leverage RIM with an approximate label prior to improve the semi-supervised learning.

3. We demonstrate the efficiency of our method in 4 challenging MuJoCo environments, showcasing that even in scenarios with highly imbalanced demonstrations, the policy can still reproduce the desired behavior modes.

## 2   Related Work

**Generative adversarial imitation learning**   GAIL adapts techniques developed for Generative Adversarial Networks (GANs) [19], and has demonstrated promising results in challenging imitation

learning tasks, such as robot control [10], imitation from raw visual data [20], multi-agent gaming [9], text generation [21], and driver behavior imitation [22]. Subsequent research has proposed various improvements to GAIL [23, 24], resulting in more efficient and reliable imitation results. However, discovering rich disentangled factors from an unstructured dataset using GAIL can be challenging due to the absence of inductive biases on the data or model. One promising approach to alleviate this issue is to introduce a latent variable model and use the technique of maximizing the mutual information [25] between the latent variable and the data, which can enable effective disentanglement of behaviors.

**Learning multi-modal behaviors**   Tasks can often be accomplished in multiple ways, and the expert may exhibit different modes of behavior [26, 27]. Therefore, it is crucial to extend GAIL to learn and replicate these multi-modal behaviors. One intuitive approach is to train separate discriminators for each labeled mode [28]. Or one can use an auxiliary classifier allowing label-conditional imitation learning about multiple intentions [15]. These supervised approaches have demonstrated their effectiveness and reliability in various applications, from controlling wheeled robots [28] to task allocation in taxi companies [29]. However, they are constrained by their reliance on labeled data. To overcome this limitation, recent research has focused on unsupervised methods that typically leverage mutual information between latent variables and the generated data [30, 12]. For example, InfoGAIL maximizes this mutual information to learn from visual demonstrations [13], while Burn-InfoGAIL maximizes it from the perspective of Bayesian inference to draw modal variables from burn-in demonstrations [31]. However, unsupervised methods often struggle to distinguish latent factors without considering semantic information or task context [32], and proved unreliable in theory [14]. Therefore, we propose a novel semi-supervised approach that leverages the advantages of both supervised and unsupervised methods to learn semantically interpretable behavior representations with a small portion of labeled data.

**Learning from imbalanced data**   Real-world data often exhibits imbalanced distributions, which can impede learning by causing the model to become biased towards the dominant category and potentially overlook the minority category. In a supervised learning setting, the problem of imbalanced data can be mitigated through various techniques such as data re-sampling and category re-weighting [33, 34]. These methods are useful when the category distributions are known beforehand. Several methods have been proposed for the unsupervised setting to deal with the imbalanced data with unknown category distributions [35, 36, 37]. A recent approach called Elastic-InfoGAN [38] has been proposed to address the issue of distribution alignment in GANs by utilizing a learnable latent distribution to handle imbalanced data. Unlike these methods, our work focuses on multi-modal semi-supervised imitation learning from raw imbalanced data without relying on prior knowledge of category distributions.

## 3 Approach

In this part, we first provide a brief background on RL, GAIL, and InfoGAIL, then provide a detailed description of how to extend InfoGAIL to a semi-supervised architecture and address the disentangled behavior learning using both the imbalanced unlabeled expert demonstrations and limited labeled expert demonstrations.

### 3.1 Background

**Reinforcement learning**   We consider a standard Markov Decision Process (MDP) setting [1] represented by a tuple $\langle \mathcal{S}, \mathcal{A}, \mathcal{P}, \mathcal{R}, \gamma \rangle$, where $\mathcal{S}$ is the state space, $\mathcal{A}$ is the action space, $\mathcal{P}\left(\mathbf{s}_{t+1} | \mathbf{s}_t, \mathbf{a}_t\right)$ is the transition probability of state $\mathbf{s}_{t+1}$ at time step $t + 1$ given state $\mathbf{s}_t$ and action $\mathbf{a}_t$ at time step $t$, $\mathcal{R}(\mathbf{s}, \mathbf{a})$ is the reward function, and $\gamma \in (0, 1)$ is the discount factor. In an RL process, an agent interacts with the environment and learns a policy $\pi(\mathbf{a} | \mathbf{s})$ to maximize the expected discounted return $\mathbb{E}_{p(\tau | \pi)}\left[\sum_{t=0}^{T-1} \gamma^t \mathcal{R}\left(\mathbf{s}_t, \mathbf{a}_t\right)\right]$, where $p(\tau | \pi) = p(\mathbf{s}_0) \prod_{t=0}^{T-1} p\left(\mathbf{s}_{t+1} | \mathbf{s}_t, \mathbf{a}_t\right) \pi(\mathbf{a}_t | \mathbf{s}_t)$ represents the likelihood of a trajectory $\tau = \{\mathbf{s}_0, \mathbf{a}_0, r_0, \mathbf{s}_1, \ldots, \mathbf{s}_{T-1}, \mathbf{a}_{T-1}, r_{T-1}, \mathbf{s}_T\}$ under $\pi$. The reward function serves as an interface through which users can specify the task that an agent needs to perform. However, creating effective reward functions that elicit the desired behaviors from an agent can often be a laborious and challenging process.

**GAIL and InfoGAIL** As one of the apprenticeship learning method, GAIL adapts techniques developed for GANs to directly learn a stylized policy through expert demonstrations without the need for handcrafted rewards. The formal GAIL objective is denoted as

$$\min_{\pi} \max_{D} V_{\text{GAIL}}(\pi, D) = \mathbb{E}_{d^{\pi}(\mathbf{s},\mathbf{a})}[\log D(\mathbf{s},\mathbf{a})] + \mathbb{E}_{d^{E}(\mathbf{s},\mathbf{a})}[\log(1 - D(\mathbf{s},\mathbf{a}))] - H(\pi), \quad (1)$$

where $D$ is a discriminator which tries to distinguish state-action pairs $(\mathbf{s}, \mathbf{a})$ drawn from the generated state-action distribution $d^{\pi}(\mathbf{s}, \mathbf{a})$, induced by $\pi$, and the state-action distribution of the expert demonstration $d^{E}(\mathbf{s}, \mathbf{a})$. $H(\pi)$ is the causal entropy of the policy $\pi$. The GAIL objective is optimized alternately by taking a gradient step to increase Equation 1 with respect to $D$ and an RL step to decrease Equation 1 with respect to $\pi$. The Jensen-Shannon divergence between $d^{\pi}(\mathbf{s}, \mathbf{a})$ and $d^{E}(\mathbf{s}, \mathbf{a})$ is eventually minimized [39].

GAIL may encounter difficulties in learning multi-modal behaviors from unstructured demonstrations in the absence of meaningful representations. To overcome this limitation, InfoGAIL introduces an additional regularization term to maximize the mutual information $I(\mathbf{c}; \mathbf{s}, \mathbf{a})$ between the latent variable $\mathbf{c}$ and state-action transitions $(\mathbf{s}, \mathbf{a})$, and the objective is then approximated by a variational lower bound [25]:

$$L_{\text{I}}(\pi, Q) = \mathbb{E}_{p(\mathbf{c}), d^{\pi}(\mathbf{s},\mathbf{a}|\mathbf{c})}[\log Q(\mathbf{c}|\mathbf{s},\mathbf{a})] + H(\mathbf{c}) \leq I(\mathbf{c}; \mathbf{s}, \mathbf{a}), \quad (2)$$

where $Q(\mathbf{c}|\mathbf{s}, \mathbf{a})$ is an approximation of the true posterior $P(\mathbf{c}|\mathbf{s}, \mathbf{a})$. The lower bound is tight if $Q = P$. This regularization allows InfoGAIL infer the interpretable latent structure in an unsupervised manner, and the final objective becomes:

$$\min_{\pi, Q} \max_{D} V_{\text{InfoGAIL}}(\pi, D, Q) = V_{\text{GAIL}}(\pi, D) - \lambda_1 L_{\text{I}}(\pi, Q), \quad (3)$$

where $\lambda_1$ is a weighting coefficient.

## 3.2 Semi-supervised Imitation Learning from Imbalanced Demonstrations

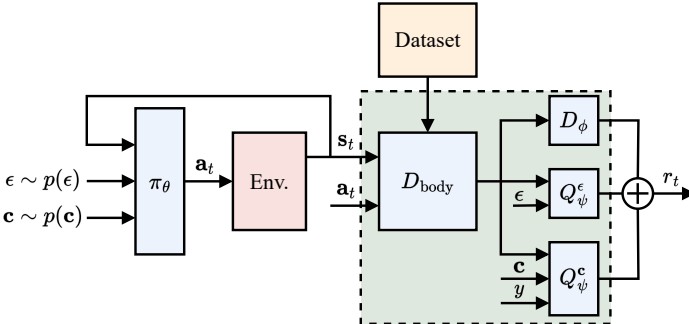

Figure 2: Schematic overview of the Ess-InfoGAIL network architecture. Here, $\mathbf{c}$ represents a latent skill variable sampled from the differentiable Gumbel-Softmax distribution, $\epsilon$ denotes a latent shifting variable sampled from a continuous uniform distribution, and $y$ signifies the category label. $D_{\text{body}}$ serves as a shared backbone network for extracting features from state-action pairs. Parameters $\theta$, $\phi$, and $\psi$ correspond to the policy $\pi_\theta$, the discriminator $D_\phi$, and the encoders $Q_\psi^\epsilon$, $Q_\psi^\mathbf{c}$, respectively.

As InfoGAIL is learned in an unsupervised manner, obtaining the desired disentangled behavior representations can prove to be challenging. The difficulty is compounded when the expert demonstrations are imbalanced, as shown in Fig. 1. To address this, we propose Elastic semi-supervised InfoGAIL (Ess-InfoGAIL) with three improvements to InfoGAIL: i) A semi-supervised learning architecture, ii) a learnable latent skill distribution and iii) RIM with an approximate label prior. The network architecture of Ess-InfoGAIL is shown in Fig. 2.

**Semi-supervised InfoGAIL** To utilize both labeled and unlabeled data, we decompose the latent variable into a semi-supervised part $\mathbf{c}$ and an unsupervised part $\epsilon$, inspired by ss-InfoGAN [16]. Unlike ss-InfoGAN, here we focus on imitation learning under sequential decision-making tasks. Additionally, we use $\mathbf{c}$ as a latent skill variable which is sampled from a categorical distribution,

allowing for encoding the same information as the label $y$, and use $\epsilon$ as a latent shifting variable which is sampled from a continuous uniform distribution, allowing for style shifting within a given skill.

Then we seek to maximize two mutual information terms $I(\epsilon; \mathbf{s}, \mathbf{a})$ and $I(\mathbf{c}; \mathbf{s}, \mathbf{a})$, where the first term can be approximated using Equation 2 with a posterior approximation $Q_\psi^\epsilon(\epsilon|\mathbf{s}, \mathbf{a})$, and the second term can be approximated using the following variational lower bounds:

$$L_{\text{IS}}^1(Q_\psi^1) = \mathbb{E}_{p(y), d^{EL}(\mathbf{s},\mathbf{a}|y)}[\log Q_\psi^1(y|\mathbf{s}, \mathbf{a})] + H(y) \leq I(y; \mathbf{s}, \mathbf{a}), \quad (4)$$

$$L_{\text{IS}}^2(\pi, Q_\psi^2) = \mathbb{E}_{p(\mathbf{c}), d^\pi(\mathbf{s},\mathbf{a}|\mathbf{c})}[\log Q_\psi^2(\mathbf{c}|\mathbf{s}, \mathbf{a})] + H(\mathbf{c}) \leq I(\mathbf{c}; \mathbf{s}, \mathbf{a}), \quad (5)$$

where, Equation 4 is a supervised term that uses the state-action distribution of the labeled expert demonstration $d^{EL}(\mathbf{s}, \mathbf{a}, y)$ to predict $y$, and Equation 5 is an unsupervised term that utilizes the state-action distribution produced by a policy $\pi_\theta$ to learn the inherent semantic meaning of $y$. The technique of lower bounding mutual information is known as variational information maximization [25]. With $Q_\psi^1 = Q_\psi^2 = Q_\psi^\mathbf{c}$ we obtain the semi-supervised regularization term:

$$L_{\text{IS}}(\pi_\theta, Q_\psi^\mathbf{c}) = L_{\text{IS}}^1(Q_\psi^\mathbf{c}) + L_{\text{IS}}^2(\pi_\theta, Q_\psi^\mathbf{c}), \quad (6)$$

where, $L_{\text{IS}}^1$ is optimized with respect to $Q_\psi^\mathbf{c}$, and $L_{\text{IS}}^2$ is optimized with respect to $\pi_\theta$. Specifically, $L_{\text{IS}}$ is trained as a classification process with the cross-entropy function, and $L_{\text{I}}$ in Equation 2 is trained as a regression process with the mean absolute error.

**Learnable latent skill distribution**  The majority of GAN and GAIL algorithms assume that the latent variable is drawn from a uniform distribution with fixed parameters. However, in the presence of imbalanced data, this assumption may lead to suboptimal optimization results. To align the state-action distribution produced by a policy $\pi_\theta$ with the state-action distribution of the imbalanced expert demonstration, we utilize a latent skill variable $\mathbf{c}$ drawn from a differentiable Gumbel-Softmax distribution $p(\mathbf{c})$ [17], similar to Elastic-InfoGAN [38]. The Gumbel-Softmax distribution is a continuous distribution over the simplex that can approximate samples from a categorical distribution in a differentiable way, and this trick has been widely used in many areas [40, 41].

Let $p_1, p_2, \cdots, p_K$ be the class probabilities of a categorical distribution, the sampling of a K-dimensional latent skill variable $\mathbf{c}$ can be done as

$$c_i = \frac{\exp\left((\log(p_i) + g_i)/\tau\right)}{\sum_{j=1}^K \exp\left((\log(p_j) + g_j)/\tau\right)} \quad \text{for } i = 1, \ldots, K, \quad (7)$$

where $g_1 \cdots g_K$ are i.i.d samples drawn from Gumbel(0, 1). The coefficient $\tau$ is a temperature parameter that determines the degree of smoothing applied to the probability distribution. A smaller value of $\tau$ leads to a distribution that is closer to the one-hot encoding, while a larger value of $\tau$ results in a distribution that is closer to the categorical distribution.

Since the objective of GAIL is to minimize the Jensen-Shannon divergence between $d^\pi(\mathbf{s}, \mathbf{a})$ and $d^E(\mathbf{s}, \mathbf{a})$, we can eventually estimate the label distribution of the unlabeled expert demonstrations by passing the gradient back to the latent skill distribution $p(\mathbf{c})$ using Gumbel-Softmax reparameterization trick.

**RIM with an approximate label prior**  Discriminative clustering techniques can automatically identify the boundaries or distinctions between categories in unlabeled data [42]. To leverage the intrinsic information in the unlabeled imbalanced demonstrations and enhance the efficiency of the semi-supervised learning process in Equation 6, we employ Regularized Information Maximization (RIM) [18] with a label prior that approximates the learned latent skill distribution:

$$L_{\text{RIM}}(Q_\psi^\mathbf{c}, p(\mathbf{c})) = \underbrace{-\frac{1}{N}\sum_i H\left(Q_\psi^\mathbf{c}(\hat{y}|\mathbf{s}_i, \mathbf{a}_i)\right)}_{1} - \underbrace{D_{KL}\left(Q_\psi^\mathbf{c}(\hat{y})\|p(\mathbf{c})\right)}_{2} - \underbrace{R(\psi)}_{3}, \quad (8)$$

where the first term is the cluster assumption [43], corresponding to the idea that datapoints should be classified with large margin. The second term is performed to avoid degenerate solutions by minimizing the KL divergence between the empirical label distribution $Q_\psi^\mathbf{c}(\hat{y}) \approx \frac{1}{N}\sum_i Q_\psi^\mathbf{c}(\hat{y}|\mathbf{s}_i, \mathbf{a}_i)$ and the learned latent skill distribution $p(\mathbf{c})$, while the last term is a parameter regularization (e.g., L2 regularization) performed to avoid complex solutions.

### 3.3 Overall Algorithm

With the above three improvements, the objective function of Ess-InfoGAIL eventually becomes:

$$\min_{\pi_\theta, Q_\psi^\epsilon, Q_\psi^{\mathbf{c}}, p(\mathbf{c})} \max_{D_\phi} V_{\text{Ess-InfoGAIL}}(\pi_\theta, D_\phi, Q_\psi^\epsilon, Q_\psi^{\mathbf{c}}, p(\mathbf{c}))$$
$$= V_{\text{InfoGAIL}}(\pi_\theta, D_\phi, Q_\psi^\epsilon, p(\mathbf{c})) - \lambda_2 L_{\text{IS}}(\pi_\theta, Q_\psi^{\mathbf{c}}, p(\mathbf{c})) - \lambda_3 L_{\text{RIM}}(Q_\psi^{\mathbf{c}}, p(\mathbf{c})), \quad (9)$$

where $\lambda_2$ and $\lambda_3$ are two weighting coefficients. $\theta, \phi, \psi$ are parameters for the policy $\pi_\theta$, the discriminator $D_\phi$, and the encoders $Q_\psi^\epsilon$ and $Q_\psi^{\mathbf{c}}$, respectively. We optimize $D_\phi$, $Q_\psi^\epsilon$ and $Q_\psi^{\mathbf{c}}$ with the stochastic gradient method. $\pi_\theta$ is optimized using Proximal Policy Optimization (PPO) [44], with advantages computed using $GAE(\lambda)$ [45], and the value function is updated using $TD(\lambda)$ [1]. The latent skill variable $\mathbf{c}$, concatenated with the state, serves as an input to the policy, where $\mathbf{c}$ is sampled from the learnable latent skill distribution $p(\mathbf{c})$ which is updated along with the policy. The schematic overview of the Ess-InfoGAIL network architecture is shown in Fig. 2, and the pseudo code is shown in Algorithm 1. In the experiments, we show that the proposed Ess-InfoGAIL can learn disentangled behavior representations efficiently from imbalanced demonstrations using limited labeled data.

---

**Algorithm 1:** Ess-InfoGAIL

    **Input:** Unlabeled and labeled demonstrations $d^E(\mathbf{s}, \mathbf{a})$, $d^{EL}(\mathbf{s}, \mathbf{a}, y)$, initial parameters of
            the policy, value function, discriminator and encoders $\theta_0, \beta_0, \phi_0, \psi_0$, initial latent
            skill distribution $p_0(\mathbf{c})$, weighting coefficients $\lambda_1, \lambda_2, \lambda_3$
    **Output:** Learned policy $\pi_{\theta*}$

1  **for** $i = 0, 1, 2, \cdots$ **do**
2      Sample a batch of latent variables $\epsilon \sim p(\epsilon), \mathbf{c} \sim p_i(\mathbf{c})$
3      Interact with the environment using $\pi_{\theta_i}, \epsilon$ and $\mathbf{c}$, and obtain $d^\pi(\mathbf{s}, \mathbf{a})$
4      Sample a batch of state-action pairs $b^\pi \sim d^\pi, b^E \sim d^E, b^{EL} \sim d^{EL}$
5      Update $\phi_i$ to $\phi_{i+1}$ by ascending with gradients according to $V_{\text{GAIL}}$
6      Update $\psi_i$ to $\psi_{i+1}$ by descending with gradients according to $V_{\text{Ess-InfoGAIL}}$
7      Update $\theta_i, p_i(\mathbf{c})$ to $\theta_{i+1}, p_{i+1}(\mathbf{c})$ by taking a policy step using the PPO update rule
         according to $V_{\text{Ess-InfoGAIL}}$
8      Update the value function using $TD(\lambda)$
9  **end**

---

## 4 Experiments

In this section, we validate our method by conducting experiments in a variety of environments, including a simple 2D trajectory environment, as well as 4 challenging MuJoCo environments (Reacher, Pusher, Walker-2D, and Humanoid). First, we perform the quantitative analysis to demonstrate the superiority of our method in discovering disentangled behavior representations from imbalanced demonstrations with limited labeled data, as compared to baseline methods. Second, we analyze the amount of labeled data required for the model to effectively encode the semantic meaning of the labels. Finally, we analyze the effect of varying the degree of data imbalance and the number of behavior modes on the imitation of multi-modal behaviors.

**Task setup** **(1) 2D trajectory** environment (Fig. 1) is a plane where the agent can move freely at a constant velocity by selecting its direction $\mathbf{p}_t$ at discrete time t. The agent's observations at time $t$ consist of positions from $t-4$ to $t$. The expert demonstrations consist of 4 distinct modes, each produced by a stochastic expert policy that generates a circular trajectory. The setting is similar to the experiment in [13]. **(2) Reacher** environment (Fig. 3 *Left*) consists a 2-DoF robot arm. The goal is to move the robot's end effector close to a target position. The original environment is modified by setting 6 evenly distributed targets on a circle with a radius of 1.5, and the target position is removed from the observation space. The initial angles of the two joints are randomly sampled between $[-\pi, \pi]$ and $[-\pi/2, \pi/2]$, respectively. **(3) Pusher** environment (Fig. 3 *Right*) consists a 4-DoF robot arm. The goal is to move a target cylinder to a goal position using the robot's end effector. Similar to Reacher, we use 6 evenly distributed goal positions and remove the goal position from the observation space. **(4) Walker-2D** environment (Fig. 4 *Left*) is a 6-DoF bipedal robot consisting of two legs

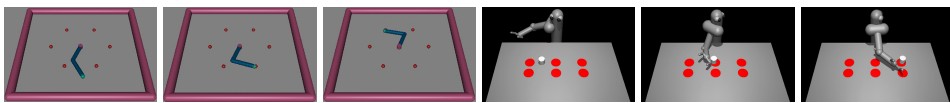

Figure 3: *Left*: Reacher with 6 targets: random initial state, reaching one target, reaching another target. *Right*: Pusher with 6 goal positions: random initial state, reaching one goal position, reaching another goal position.

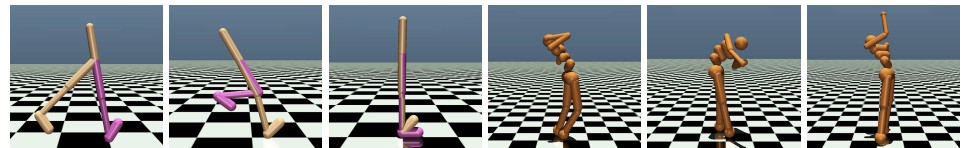

Figure 4: *Left*: Walker-2D with 3 goals: running forwards, running backwards, balancing. *Right*: Humanoid with 3 goals: running forwards, running backwards, balancing.

and feet attached to a common base. The goal is to make coordinate both sets of legs and feet to move the robot in the right direction, including running forward, running backward and balancing. **(5) Humanoid** environment (Fig. 4 *Right*) is a high-dimensional robot with 17 degrees of freedom. The goal is similar to the Walker-2D, including running forward, running backward and balancing.

**Multi-modal demonstrations preparation**    In the experiments, we first pre-train K expert policies, each corresponding to K different goals (or K behavior modes). Subsequently, we use these K expert policies to sample K sets of expert demonstrations. From each set of expert demonstrations, we extract a small portion and label them with the one-hot behavioral categories, while the remaining expert demonstrations are randomly sampled and mixed to create imbalanced unlabeled expert data. Moreover, in the real-world scenario, one can directly use the raw motion capture data (e.g., motion capture data of an animal over a day) without the need to train additional expert policies.

**Baselines and evaluation metrics**    We design various baselines to demonstrate the efficiency of our method in learning multi-modal behaviors from imbalanced demonstrations. **(1) GAIL**: The original GAIL. **(2) InfoGAIL**: The original InfoGAIL with a fixed uniform categorical distribution. **(3) ACGAIL**: Reproduction of ACGAIL [15] using an auxiliary classifier with full labeled data. **(4) Elastic-InfoGAIL**: Modification of Elastic-InfoGAN [38] in imitation tasks with a learnable latent distribution and a contrastive loss. **(5) Ess-InfoGAIL\GS**: Ess-InfoGAIL without using the Gumbel-Softmax technique. **(6) Ess-InfoGAIL\RIM**: Ess-InfoGAIL without using the RIM technique. **(7) Ess-InfoGAIL**: Our final method.

To evaluate the behavior disentanglement quality of the above methods, we use a pre-trained behavior classifier and employ two commonly used metrics in clustering algorithms similar to [38]: **(a) Average Entropy (ENT)** evaluates two properties: i) whether the state-action pairs generated for a given latent variable belong to the same ground-truth class and ii) whether each ground-truth class is associated with a single unique latent variable. **(b) Normalized Mutual Information (NMI)** quantifies the correlation between two clusterings, with a value ranging from 0 to 1. A higher NMI value indicates a stronger correlation between the clusterings.

Additionally, we use the normalized average task reward to assess the quality of learned policies. Note that, we modify the original MuJoCo environment and compute the task reward differently for each behavior mode. The final task reward, derived by averaging across all modes, serves exclusively as an evaluation metric and does not affect the policy training process.

## 4.1   Comparison with Baseline Methods

We quantitatively analyze Ess-InfoGAIL's behavior disentanglement performance, as presented in Table 1. In complex tasks with numerous behavior modes, both GAIL and InfoGAIL often encounter mode collapse, where their policies manifest behavior corresponding to only a subset of the behavior modes due to a lack of prior behavior representation knowledge. For instance, in the Reacher environment, the robot arm reaches only a few target positions. Leveraging limited labeling

Table 1: Behavior disentanglement quality measured by NMI (↑) and ENT (↓). Where Ess-InfoGAIL\GS removes Gumbel-Softmax in Ess-InfoGAIL, and Ess-InfoGAIL\RIM removes RIM in Ess-InfoGAIL. For each experiment, we use 10 different random seeds and collect 50 episodes for each seed. The results demonstrate the efficiency of Ess-InfoGAIL in learning multi-modal behaviors compared to baseline methods. (See supplementary for error bars.)

| | 2D trajectory | | Reacher | | Pusher | | Walker-2D | | Humanoid | |
| --- | --- | --- | --- | --- | --- | --- | --- | --- | --- | --- |
| Metrics | NMI | ENT | NMI | ENT | NMI | ENT | NMI | ENT | NMI | ENT |
| GAIL [8] | 0.392 | 0.529 | 0.153 | 1.055 | 0.376 | 0.723 | 0.439 | 0.406 | 0.388 | 0.597 |
| InfoGAIL [13] | 0.742 | 0.371 | 0.301 | 1.113 | 0.604 | 0.551 | 0.657 | 0.284 | 0.550 | 0.487 |
| ACGAIL [15] | 0.783 | 0.324 | 0.537 | 0.781 | 0.754 | 0.409 | 0.658 | 0.340 | 0.544 | 0.478 |
| Elastic-InfoGAIL | 0.773 | 0.330 | 0.311 | 1.101 | 0.650 | 0.537 | 0.615 | 0.351 | 0.503 | 0.498 |
| Ess-InfoGAIL\GS | 0.892 | 0.157 | 0.607 | 0.661 | 0.857 | 0.194 | 0.740 | 0.262 | 0.638 | 0.360 |
| Ess-InfoGAIL\RIM | 0.893 | 0.159 | 0.575 | 0.725 | 0.875 | 0.188 | 0.715 | 0.274 | 0.642 | 0.347 |
| Ess-InfoGAIL (Ours) | **0.910** | **0.131** | **0.662** | **0.587** | **0.906** | **0.144** | **0.755** | **0.237** | **0.696** | **0.206** |

guidance, our approach, Ess-InfoGAIL, improves the average NMI by 0.436 and 0.215 and decreases the average ENT by 0.401 and 0.300 compared to GAIL and InfoGAIL, respectively, across all environments. Notably, Ess-InfoGAIL attains superior behavior disentanglement quality with fewer labels than ACGAIL, which utilizes full labeled data. This success might stem from imbalanced data hindering ACGAIL's auxiliary classifier learning, while our method balances the small number of labels by extracting equal-duration data for each behavior mode.

Furthermore, we adapt Elastic-InfoGAN to the imitation learning framework using Gumbel-Softmax and contrastive learning loss, augmenting negative samples with Gaussian noise. However, contrastive learning's efficacy is limited in visual-free imitation tasks, and it indirectly addresses behavior disentanglement, yielding uncertain performance. In contrast, Ess-InfoGAIL directly tackles mode collapse and achieves semantically interpretable behavior disentanglement via limited labels. It exhibits an average NMI improvement of 0.215 and an average ENT decrease of 0.302 over Elastic-InfoGAIL across all environments. We perform ablation experiments, confirming the enhancements brought by Gumbel-Softmax and RIM techniques under the semi-supervised framework and imbalanced data. Ess-InfoGAIL demonstrates an average NMI increase of 0.039 and 0.046 and an average ENT reduction of 0.066 and 0.078 compared to Ess-InfoGAIL\GS and Ess-InfoGAIL\RIM, respectively.

To assess the quality of learned policies, we analyze the normalized average task rewards within the Reacher environment, depicted in Fig. 5, demonstrating the superiority of our method over all baseline methods. Results for different environments can be found in the supplementary material. Due to mode collapse and data imbalance, some methods (e.g., GAIL and InfoGAIL) may end up learning only one or two behavior modes, leading to lower average task rewards compared to a random policy.

## 4.2 Amount of Labeled Data Requirements

In real-world scenarios, obtaining labeled behavior data (e.g., from motion capture or raw video) is difficult due to the interleaving of various behavior modes. Thus, minimizing labeling costs is

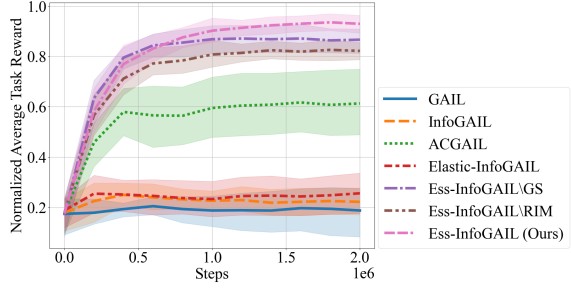

Figure 5: Normalized average task reward of each method during training (only for evaluation).

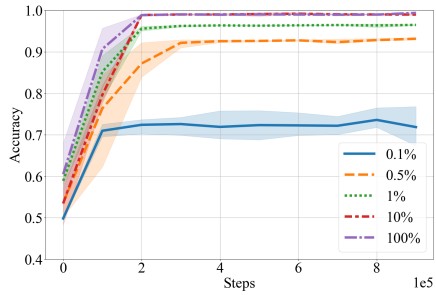

Figure 6: Classification accuracy using different amount of labeled data.

crucial. We assess the prediction accuracy of encoders trained with limited labeled data across varying proportions within all unlabeled data. Results are shown in Fig. 6 (using Reacher environment as an example). Proportions of labeled data are set at 0.1%, 0.5%, 1%, 10%, and 100%. For Reacher, 0.1% data equates to 2 episodes (100 time steps) per mode. The encoder demonstrates nearly 92% prediction accuracy with just 0.5% labeled data. Walker-2D and Humanoid tasks, with higher state dimensions, require more labeled data for effective behavior mode separation. Default label ratios, if not specified, are: 2D-Trajectory: 1%, Reacher: 0.5%, Pusher: 1%, Walker-2D: 1%, Humanoid: 2%. Guided by limited labeled data, the agent adeptly disentangles specific styles from labeled data and more varied behavioral styles from unlabeled data.

Table 2: Degree of data imbalance.

| | InfoGAIL | | Ess-InfoGAIL | |
|---|---|---|---|---|
| Metrics | NMI | ENT | NMI | ENT |
| 20 | 0.328 | 1.035 | 0.704 | 0.510 |
| 40 | 0.317 | 1.050 | 0.693 | 0.521 |
| 60 | 0.313 | 1.061 | 0.687 | 0.583 |
| 80 | 0.306 | 1.073 | 0.681 | 0.561 |
| 100 | 0.301 | 1.113 | 0.662 | 0.587 |
| 200 | 0.291 | 1.121 | 0.625 | 0.607 |

Table 3: Learning more behavior modes.

| | InfoGAIL | | Ess-InfoGAIL | |
|---|---|---|---|---|
| Metrics | NMI | ENT | NMI | ENT |
| 2 | 0.329 | 1.031 | 0.711 | 0.504 |
| 4 | 0.319 | 1.073 | 0.703 | 0.522 |
| 6 | 0.301 | 1.113 | 0.662 | 0.587 |
| 8 | 0.308 | 1.087 | 0.651 | 0.619 |
| 10 | 0.303 | 1.076 | 0.617 | 0.657 |
| 12 | 0.298 | 1.115 | 0.613 | 0.653 |

## 4.3 Degree of Data Imbalance

The degree of data imbalance directly affects the learning quality of the policy, discriminator, and encoder. We set up 6 levels of data imbalance in the Reacher environment, ranging from 20 to 200. For example, at a data imbalance level of 20, the majority class has at most 20 times the number of samples as the minority class. The proportions of labeled data is set to 0.5%. At the maximum data imbalance level of 200, the minority class has only about 2 episodes of trajectories for each mode. The experimental results are shown in Table 2. It can be observed that as the degree of data imbalance increases, the NMI gradually decreases while the ENT gradually increases. However, even when the degree of the data imbalance is 200, the NMI of Ess-InfoGAIL only decreases by 0.079 compared to the case of a data imbalance of 20, and the performance is still better than InfoGAIL with a data imbalance of 20. Our proposed method demonstrates the ability to achieve effective disentanglement of behaviors, even in the presence of highly imbalanced data distributions.

## 4.4 Learning More Behavior Modes

The disentanglement task becomes increasingly challenging with an increase in the number of behavior modes. We set up 6 levels of behavior modes ranging from 2 to 12 in the Reacher environment, where the agent needs to learn disentangled behavior representations from these unstructured behavior data. We maintain a fixed degree of data imbalance at 100 and the proportions of labeled data is set to 0.5%. The experimental results are presented in Table 3. The results demonstrate that even with 12 behavior modes, our method consistently achieves higher NMI values and lower ENT values, compared to InfoGAIL with only 2 behavior modes. This affirms the controllability and scalability of our method in learning disentangled behavior representations while preserving category semantic information.

## 5 Conclusion

In this work, we proposed a novel semi-supervised imitation learning method, Ess-InfoGAIL, which utilizes a learnable latent distribution and an improved RIM to enable the agent to learn better disentangled multi-modal behaviors from unstructured expert demonstrations. We qualitatively and quantitatively validated the efficiency of our method, compared with other baseline methods. We also demonstrated that our method only requires a very small amount of labeled data to achieve good results, and further verified its controllability and scalability in scenarios with higher imbalance degrees and more behavior modes, which are crucial for multi-modal imitation tasks in the real world. Although there are some limitations, such as the need to provide a small amount of labeled data for each category and preset the number of modal categories, we believe that our work lays the foundation for imitation learning in large-scale unstructured demonstrations.

## Acknowledgments

This work was supported in part by the National Natural Science Foundation of China (No. 62073160).

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
