## Supplementary

## A   Properties of the Ess-InfoGAIL

In this section, we describe some favorable properties of the proposed Ess-InfoGAIL from the perspective of information theoretic. For brevity, let $(\mathbf{s}, \mathbf{a})$, $(\tilde{\mathbf{s}}, \tilde{\mathbf{a}}) \in \mathbf{x}$ indicate the expert state-action pair and the generated state-action pair, respectively, and $y, \mathbf{c} \in \mathbf{z}$ indicate the ground truth label and the latent variable, respectively.

Firstly, by increasing $I(y; \mathbf{s}, \mathbf{a})$ and $I(\mathbf{c}; \tilde{\mathbf{s}}, \tilde{\mathbf{a}})$, both $I(y; \mathbf{c})$ and $I(\mathbf{s}, \mathbf{a}; \tilde{\mathbf{s}}, \tilde{\mathbf{a}})$ are increased as well. To prove this, we introduce two conditional independence assumptions: i) given $\mathbf{x}$, $y$ and $\mathbf{c}$ are independent, ii) given $\mathbf{z}$, $(\mathbf{s}, \mathbf{a})$ and $(\tilde{\mathbf{s}}, \tilde{\mathbf{a}})$ are independent. The multivariate mutual information $I(\mathbf{x}; y; \mathbf{c})$ can be decomposed as

$$
\begin{aligned}
I(\mathbf{x}; y; \mathbf{c}) &= I(y; \mathbf{x}) + I(\mathbf{c}; \mathbf{x}) - I(y, \mathbf{c}; \mathbf{x}) \\
&= I(y; \mathbf{x}) + I(\mathbf{c}; \mathbf{x}) - H(y, \mathbf{c}) + H(y, \mathbf{c}|\mathbf{x}) \\
&= I(y; \mathbf{c}) - I(y; \mathbf{c}|\mathbf{x}).
\end{aligned}
\tag{10}
$$

According to assumption i), the last two lines of Equation 10 can be transformed as

$$
H(y, \mathbf{c}) = I(y; \mathbf{x}) + I(\mathbf{c}; \mathbf{x}) + H(y, \mathbf{c}|\mathbf{x}) - I(y; \mathbf{c}),
\tag{11}
$$

where, $H(y, \mathbf{c})$ can be regarded as a constant. Let $\Delta$ represent the amount of change for each term of Equation 11, then we have

$$
\Delta_{I(y;\mathbf{x})} + \Delta_{I(\mathbf{c};\mathbf{x})} + \Delta_{H(y,\mathbf{c}|\mathbf{x})} - \Delta_{I(y;\mathbf{c})} = 0.
\tag{12}
$$

By directly increasing $I(y; \mathbf{x})$ and $I(\mathbf{c}; \mathbf{x})$ during training, the following two unequal relationships will hold:

$$
\begin{aligned}
\Delta_{I(y;\mathbf{x})} + \Delta_{I(\mathbf{c};\mathbf{x})} \geq -\Delta_{H(y,\mathbf{c}|\mathbf{x})} \Longrightarrow \Delta_{I(y;\mathbf{c})} \geq 0 \\
\Delta_{I(y;\mathbf{x})} + \Delta_{I(\mathbf{c};\mathbf{x})} < -\Delta_{H(y,\mathbf{c}|\mathbf{x})} \Longrightarrow \Delta_{I(y;\mathbf{c})} < 0
\end{aligned}
\tag{13}
$$

According to the property of mutual information, $I(y; \mathbf{c})$ cannot be negative. While the second inequality will result in a negative $I(y; \mathbf{c})$, which violates the property. Thus, by increasing $I(y; \mathbf{s}, \mathbf{a})$ and $I(\mathbf{c}; \tilde{\mathbf{s}}, \tilde{\mathbf{a}})$, $I(y; \mathbf{c})$ is also increased. Similarly, by using the assumption ii), it can be proved that $I(\mathbf{s}, \mathbf{a}; \tilde{\mathbf{s}}, \tilde{\mathbf{a}})$ is finally increased as well.

Secondly, by optimizing the RIM objective, the mutual information $I(\hat{y}; \mathbf{s}, \mathbf{a})$ is maximized, and $Q_{\psi}^{\mathbf{c}}(\hat{y})$ approaches $p(\mathbf{c})$ through cross entropy $H(Q_{\psi}^{\mathbf{c}}(\hat{y}) \| p(\mathbf{c}))$:

$$
\begin{aligned}
L_{\text{RIM}}(Q_{\psi}^{\mathbf{c}}) &= -\frac{1}{N} \sum_i H\left(Q_{\psi}^{\mathbf{c}}(\hat{y}|\mathbf{s}_i, \mathbf{a}_i)\right) - D_{KL}\left(Q_{\psi}^{\mathbf{c}}(\hat{y}) \| p(\mathbf{c})\right) - R(\psi) \\
&= \mathbb{E}_{d^E(\mathbf{s},\mathbf{a})}\left[-H\left(Q_{\psi}^{\mathbf{c}}(\hat{y}|\mathbf{s}, \mathbf{a})\right) - Q_{\psi}^{\mathbf{c}}(\hat{y}) \log \frac{Q_{\psi}^{\mathbf{c}}(\hat{y})}{p(\mathbf{c})}\right] - R(\psi) \\
&= \mathbb{E}_{d^E(\mathbf{s},\mathbf{a})}\left[-H\left(Q_{\psi}^{\mathbf{c}}(\hat{y}|\mathbf{s}, \mathbf{a})\right) - Q_{\psi}^{\mathbf{c}}(\hat{y}) \log Q_{\psi}^{\mathbf{c}}(\hat{y}) + Q_{\psi}^{\mathbf{c}}(\hat{y}) \log p(\mathbf{c})\right] - R(\psi) \\
&= \mathbb{E}_{d^E(\mathbf{s},\mathbf{a})}\left[H\left(Q_{\psi}^{\mathbf{c}}(\hat{y})\right) - H\left(Q_{\psi}^{\mathbf{c}}(\hat{y}|\mathbf{s}, \mathbf{a})\right) - H\left(Q_{\psi}^{\mathbf{c}}(\hat{y}) \| p(\mathbf{c})\right)\right] - R(\psi) \\
&= \mathbb{E}_{d^E(\mathbf{s},\mathbf{a})}\left[I(\hat{y}; \mathbf{s}, \mathbf{a}) - H\left(Q_{\psi}^{\mathbf{c}}(\hat{y}) \| p(\mathbf{c})\right)\right] - R(\psi),
\end{aligned}
\tag{14}
$$

where, $p(\mathbf{c})$ is a learnable latent skill distribution that approximates the imbalanced ground truth label distribution using Gumbel-Softmax reparameterization trick.

## B   Implementation Details

The main parameters for training Ess-InfoGAIL are listed in Table 4. We use multi-layer perceptrons for our policy network, value network, discriminator and encoders, as shown in Fig. 7. The policy is parameterized as a diagonal Gaussian distribution $\pi_\theta(\mathbf{a}|\mathbf{s}, \epsilon, \mathbf{c}) \sim \mathcal{N}(\mu_{\pi_\theta}(\mathbf{s}, \epsilon, \mathbf{c}), \Sigma_{\pi_\theta})$ with a mean $\mu_{\pi_\theta}(\mathbf{s}, \epsilon, \mathbf{c})$ output from the policy network, and a state independent covariance matrix $\Sigma_{\pi_\theta}$. The value network is similar to the policy network, but with a single linear output unit. The discriminator

and encoders share network parameters with separate output layers. All networks consist of fully-connected layers with Tanh activations applied to the hidden layers, and are optimized using Adam algorithm with specific initial learning rates. To optimize the policy, we use PPO algorithm with advantages computed using $GAE(\lambda), \lambda = 0.95$.

To align the generated and expert data distributions, we use a latent skill variable sampled from a Gumbel-Softmax distribution with temperature $\tau = 0.1$. Further, to ensure more stable gradient updates for the latent skill distribution and avoid issues with values that are too small or too large, we use the logarithmic form of the latent skill distribution for gradient updates. To minimize computational time, we restrict the update of the latent skill distribution to only the first iteration of policy updates. Our experiments demonstrate that this approach does not result in significant performance degradation. The weighting coefficients $\lambda_1$ and $\lambda_2$ are set to 1.0 and 4.0 respectively. The weighting coefficient $\lambda_3$ balances the supervised classification and the unsupervised clustering of the encoders. At the beginning of training, we expect the encoder to focus more on the classification loss. Therefore, the value of $\lambda_3$ is initially set to 0. As training progresses, we gradually increase $\lambda_3$ to its maximum value of 1.0, as we want the encoder to utilize the intrinsic information in the unlabeled data and improve the performance of the classification task.

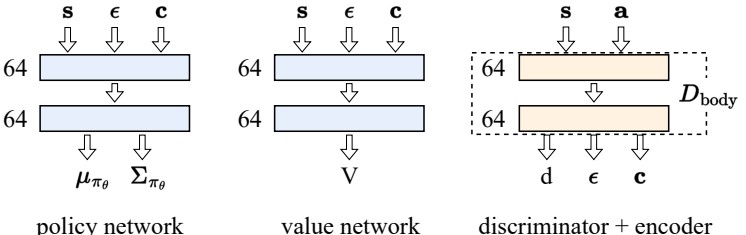

Figure 7: Network architectures used in Ess-InfoGAIL. All networks consist of fully-connected layers with Tanh activations applied to the hidden layers. The discriminator and encoder share network parameters with separate output layers.

Table 4: Parameters for training Ess-InfoGAIL.

| Parameters | Values |
| --- | --- |
| Optimizer | Adam |
| Policy/Value learning rate | 0.003 |
| Discriminator learning rate | 0.005 |
| Encoder learning rate | 0.01 |
| Discount factor | 0.99 |
| Policy/Value minibatch size | 1000 |
| Discriminator/Encoder minibatch size | 5000 |
| Policy/Value update iterations | 20 |
| Discriminator/Encoder iterations | 50 |
| PPO clip threshold | 0.2 |
| Gumbel-Softmax temperature $\tau$ | 0.1 |
| Weighting coefficient $\lambda_1$ | 1.0 |
| Weighting coefficient $\lambda_2$ | 4.0 |
| Weighting coefficient $\lambda_3$ | 3.0 |
| $GAE(\lambda)$ | 0.95 |
| $TD(\lambda)$ | 0.95 |

## C  More Experimental Results

The quantitative results from Table 1 in the main paper, along with their error bars (standard deviation), are presented in Table 5 and Table 6. Similarly, Table 7 and Table 8 display the quantitative results from Table 2 and Table 3 in the main paper, respectively, along with their corresponding error bars. It can be observed that the error bars of Ess-InfoGAIL consistently remain at a relatively low level.

As the number of behavior modes increases, the error bars slightly expand but still remain within an acceptable range. To validate the quality of the trained policies, we record the normalized average task reward during the training process, as depicted in Fig. 8. The normalized average task reward is computed across multiple random seeds and averaged over all modes. It is important to note that this reward solely serves as an evaluation metric and does not contribute to the training process. The quantified data for the normalized average task reward is presented in Table 9. All the presented outcomes showcase the advantages of our proposed Ess-InfoGAIL method.

Table 5: Behavior disentanglement quality measured by NMI ($\uparrow$).

|  | 2D trajectory | Reacher | Pusher | Walker-2D | Humanoid |
|---|---|---|---|---|---|
| GAIL [8] | 0.392± 0.087 | 0.153± 0.056 | 0.376± 0.082 | 0.439± 0.081 | 0.388± 0.067 |
| InfoGAIL [13] | 0.742± 0.045 | 0.301± 0.024 | 0.604± 0.050 | 0.657± 0.049 | 0.550± 0.045 |
| ACGAIL [15] | 0.783± 0.053 | 0.537± 0.029 | 0.754± 0.049 | 0.658± 0.065 | 0.544± 0.061 |
| Elastic-InfoGAIL | 0.773± 0.054 | 0.311± 0.029 | 0.650± 0.061 | 0.615± 0.068 | 0.503± 0.081 |
| Ess-InfoGAIL\GS | 0.892± 0.024 | 0.607± 0.033 | 0.857± 0.045 | 0.740± 0.069 | 0.638± 0.057 |
| Ess-InfoGAIL\RIM | 0.893± 0.031 | 0.575± 0.043 | 0.875± 0.030 | 0.715± 0.075 | 0.642± 0.064 |
| Ess-InfoGAIL (Ours) | **0.910± 0.035** | **0.662± 0.040** | **0.906± 0.037** | **0.755± 0.065** | **0.696± 0.050** |

Table 6: Behavior disentanglement quality measured by ENT ($\downarrow$).

|  | 2D trajectory | Reacher | Pusher | Walker-2D | Humanoid |
|---|---|---|---|---|---|
| GAIL [8] | 0.529± 0.051 | 1.055± 0.050 | 0.723± 0.092 | 0.406± 0.059 | 0.597± 0.078 |
| InfoGAIL [13] | 0.371± 0.050 | 1.113± 0.049 | 0.551± 0.060 | 0.284± 0.042 | 0.487± 0.066 |
| ACGAIL [15] | 0.324± 0.050 | 0.781± 0.047 | 0.409± 0.060 | 0.340± 0.082 | 0.478± 0.074 |
| Elastic-InfoGAIL | 0.330± 0.052 | 1.101± 0.051 | 0.537± 0.118 | 0.351± 0.080 | 0.498± 0.077 |
| Ess-InfoGAIL\GS | 0.157± 0.035 | 0.661± 0.056 | 0.194± 0.048 | 0.262± 0.066 | 0.360± 0.051 |
| Ess-InfoGAIL\RIM | 0.159± 0.050 | 0.725± 0.076 | 0.188± 0.052 | 0.274± 0.081 | 0.347± 0.056 |
| Ess-InfoGAIL (Ours) | **0.131± 0.055** | **0.587± 0.064** | **0.144± 0.055** | **0.237± 0.065** | **0.206± 0.059** |

Table 7: Degree of data imbalance.

|  | InfoGAIL | | Ess-InfoGAIL | |
|---|---|---|---|---|
| Metrics | NMI | ENT | NMI | ENT |
| 20 | 0.328± 0.022 | 1.035± 0.028 | 0.704± 0.028 | 0.510± 0.087 |
| 40 | 0.317± 0.023 | 1.050± 0.033 | 0.693± 0.047 | 0.521± 0.057 |
| 60 | 0.313± 0.023 | 1.061± 0.032 | 0.687± 0.055 | 0.583± 0.068 |
| 80 | 0.306± 0.030 | 1.073± 0.042 | 0.681± 0.043 | 0.561± 0.065 |
| 100 | 0.301± 0.024 | 1.113± 0.049 | 0.662± 0.040 | 0.587± 0.064 |
| 200 | 0.291± 0.040 | 1.121± 0.062 | 0.625± 0.048 | 0.607± 0.067 |

Table 8: Learning more behavior modes.

|  | InfoGAIL | | Ess-InfoGAIL | |
|---|---|---|---|---|
| Metrics | NMI | ENT | NMI | ENT |
| 2 | 0.329± 0.019 | 1.031± 0.029 | 0.711± 0.057 | 0.504± 0.054 |
| 4 | 0.319± 0.045 | 1.073± 0.062 | 0.703± 0.047 | 0.522± 0.077 |
| 6 | 0.301± 0.024 | 1.113± 0.049 | 0.662± 0.040 | 0.587± 0.064 |
| 8 | 0.308± 0.033 | 1.087± 0.051 | 0.651± 0.053 | 0.619± 0.089 |
| 10 | 0.303± 0.029 | 1.076± 0.062 | 0.617± 0.073 | 0.657± 0.080 |
| 12 | 0.298± 0.038 | 1.115± 0.061 | 0.613± 0.095 | 0.653± 0.078 |

Table 9: Normalized average task reward of each method (only for evaluation).

|  | 2D trajectory | Reacher | Pusher | Walker-2D | Humanoid |
|---|---|---|---|---|---|
| GAIL [8] | 0.071± 0.019 | 0.189± 0.087 | 0.338± 0.029 | 0.459± 0.014 | 0.508± 0.033 |
| InfoGAIL [13] | 0.155± 0.028 | 0.223± 0.054 | 0.431± 0.043 | 0.552± 0.059 | 0.526± 0.127 |
| ACGAIL [15] | 0.540± 0.061 | 0.616± 0.131 | 0.790± 0.042 | 0.703± 0.040 | 0.614± 0.040 |
| Elastic-InfoGAIL | 0.189± 0.059 | 0.261± 0.079 | 0.483± 0.042 | 0.616± 0.039 | 0.544± 0.111 |
| Ess-InfoGAIL\GS | 0.845± 0.023 | 0.868± 0.050 | 0.867± 0.040 | 0.812± 0.050 | 0.817± 0.017 |
| Ess-InfoGAIL\RIM | 0.882± 0.036 | 0.826± 0.035 | 0.772± 0.061 | 0.770± 0.019 | 0.751± 0.065 |
| Ess-InfoGAIL (Ours) | **0.956± 0.040** | **0.933± 0.033** | **0.967± 0.022** | **0.911± 0.042** | **0.920± 0.036** |

# D   Impact of the Latent Shifting Variable

During the training process, the discrete latent skill variable **c** dominates the behavior category representation through the semi-supervised learning, while the continuous latent shifting variable $\epsilon$

corresponds to other interpretable potential representations. We visualize the effects of adjusting $\epsilon$ within the range of -1 to 1 in the 2D trajectory environment, as shown in Fig. 9. From a column-wise perspective, (a) $\sim$ (d) correspond to 4 different behavior modes controlled by the latent skill variable $\mathbf{c}$. From a row-wise perspective, left to right correspond to continuous style variations within the same behavior mode, controlled by the latent shifting variable $\epsilon$. The representations associated with the latent shifting variable $\epsilon$ may vary depending on the task.

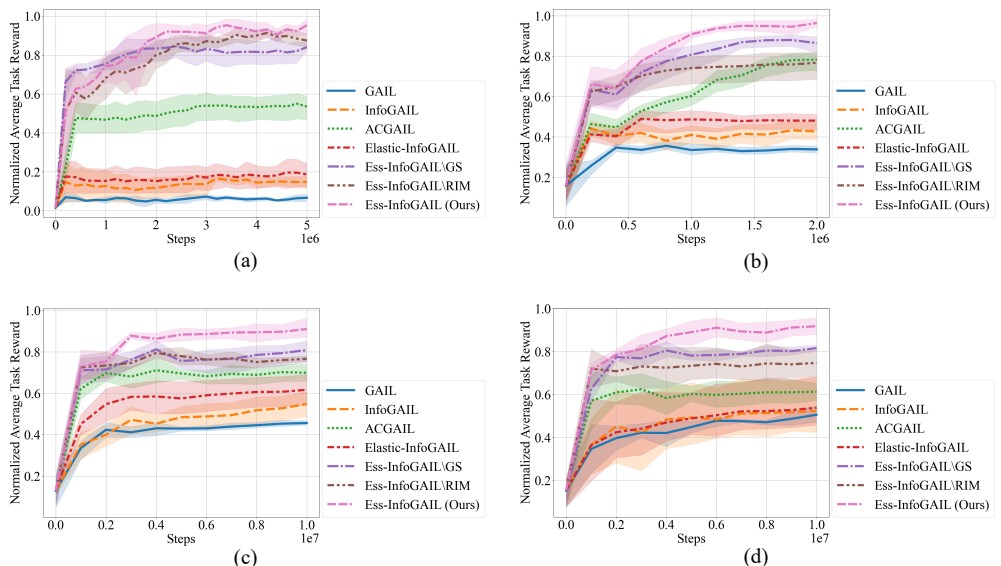

Figure 8: Normalized average task reward of each method during training (only for evaluation). (a) 2D trajectory. (b) Pusher. (c) Walker-2D. (d) Humanoid.

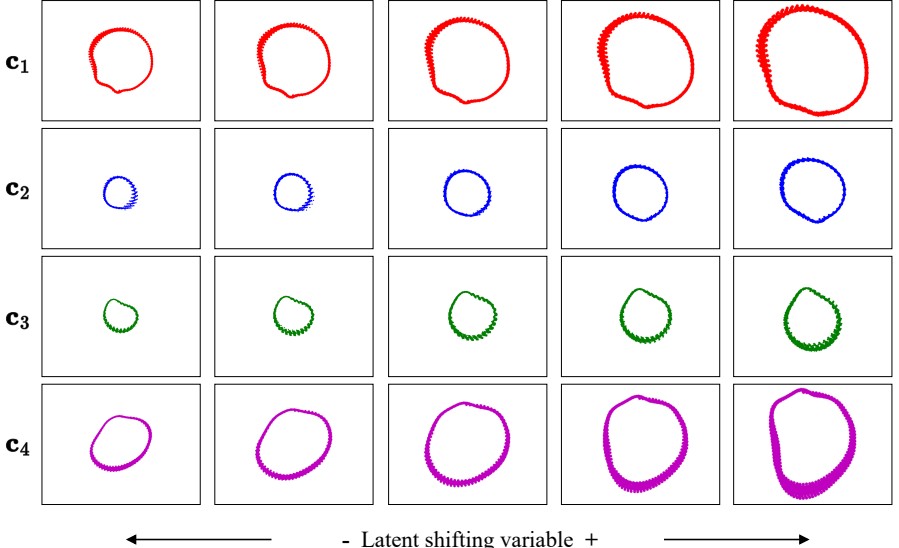

- Latent shifting variable +

Figure 9: Illustrations of modifying circle size by manipulating the latent shifting variable in the 2D trajectory environment are presented. From a column-wise perspective, $\mathbf{c}_1 \sim \mathbf{c}_4$ correspond to 4 different behavior modes controlled by the latent skill variable. From a row-wise perspective, left to right correspond to continuous style variations within the same behavior mode, controlled by the latent shifting variable $\epsilon$.

# E    Ground Truth of the Data Imbalance

Here, we describe the setting of the normalized ground truth data imbalance in different environments. For each imbalance group, we use 10 different random seeds and collect 50 episodes for each seed.

## E.1    2D trajectory

Expert demonstrations in the 2D trajectory environment consist of 4 different behavior modes, with a maximum data imbalance of 20.

- Degree of data imbalance 20: 0.05, 1.0, 1.0, 0.05

## E.2    Reacher

Expert demonstrations in the Reacher environment include from 2 to 12 different behavior modes, with a maximum data imbalance of 200.

### 2 behavior modes

- Degree of data imbalance 20: 0.05, 1.0

### 4 behavior modes

- Degree of data imbalance 40: 0.025, 0.05, 0.1, 1.0

### 6 behavior modes

- Degree of data imbalance 20: 0.05, 0.05, 0.1, 0.1, 1.0, 1.0
- Degree of data imbalance 40: 0.025, 0.025, 0.1, 0.1, 1.0, 1.0
- Degree of data imbalance 60: 0.017, 0.017, 0.1, 0.1, 1.0, 1.0
- Degree of data imbalance 80: 0.0125, 0.0125, 0.1, 0.1, 1.0, 1.0
- Degree of data imbalance 100: 0.01, 0.01, 0.1, 0.1, 1.0, 1.0
- Degree of data imbalance 200: 0.005, 0.005, 0.1, 0.1, 1.0, 1.0

### 8 behavior modes

- Degree of data imbalance 100: 0.01, 0.01, 0.1, 0.1, 0.1, 1.0, 1.0, 1.0

### 10 behavior modes

- Degree of data imbalance 100: 0.01, 0.01, 0.01, 0.1, 0.1, 0.1, 1.0, 1.0, 1.0, 1.0

### 12 behavior modes

- Degree of data imbalance 100: 0.01, 0.01, 0.01, 0.01, 0.1, 0.1, 0.1, 0.1, 1.0, 1.0, 1.0, 1.0

## E.3    Pusher

Expert demonstrations in the Pusher environment consist of 6 different behavior modes, with a maximum data imbalance of 100.

- Degree of data imbalance 100: 0.01, 0.01, 0.1, 0.1, 1.0, 1.0

## E.4    Walker-2D and Humanoid

Expert demonstrations in Walker-2D and Humanoid environments consist of 3 different behavior modes, with a maximum data imbalance of 100.

- Degree of data imbalance 100: 0.01, 0.1, 1.0