# OpenReview forum: "Ess-InfoGAIL: Semi-supervised Imitation Learning from Imbalanced Demonstrations"
_NeurIPS.cc/2023/Conference — NeurIPS 2023 poster_

### Official Review · Reviewer_dvkr · 2023-07-06

**Soundness:** 3 good
**Presentation:** 3 good
**Contribution:** 2 fair
**Rating:** 7
**Confidence:** 3

**Summary:**

The paper presents a semi-supervised approach to InfoGAIL for learning from demonstrations that enables disentangled learned behaviors. The proposed approach outperforms all considered baselines in MuJoCo simulation in imbalanced settings, and is robust to very little labeled data as well as varying levels of imbalance in the dataset.

**Strengths:**

* The theme of this paper in handling imbalanced demonstration data is timely and interesting.
* The paper is well written.
* The experiments diligently consider the potential relevant settings showcasing the capabilities of Ess-InfoGAIL in terms of handling imbalanced data, and a small fraction of labeled datapoints.

**Weaknesses:**

* There are several points of confusion for me in terms of the setting considered. From Fig. 1, it is unclear what about the expert trajectories indicates imbalance. I might have missed this, but what are the considered labels in the experiments? How many label classes are there?
* The code is not available as of yet.
* Table 1 is missing some measure of statistical significance.
* In line 263, two numbers are listed twice. Why is the improvement over both ablation variants the same for both cases?
* In Fig. 5, it seems that 0.5% labeled data achieves good performance (above 90%) on both Reacher and Pusher tasks? This observation does not seem to agree with the discussion. Could both Fig. 5(a) and 5(b) have the same y-scale to make it easier to compare?

* The following is a non-exhaustive list of errors/typos and stylistic improvements I found:
1. Line 45: 'correspond' -> 'corresponds'.
2. Lines 79-93: missing space between words and citations. In general, citations should use 'abbrv' option in the bibliography.
3. Hanging section heading (4.1).
4. The references should be proofread (e.g., IEEE should not be listed twice in a citation, appropriate words (e.g., Bayesian) should be capitalized).

**Questions:**

See above weaknesses section.

**Limitations:**

The authors mentioned some limitations for the proposed method, although this section could use more depth.

---

> ### Author Rebuttal · Authors · 2023-08-09
>
> # Response to Reviewer dvkr
>
> > Q1. From Fig. 1, it is unclear what about the expert trajectories indicates imbalance. What are the considered labels in the experiments? How many label classes are there?
>
> Fig. 1 illustrates a simple 2D-Trajectory scenario, facilitating the visual comparison of algorithm performance. In the environment, an agent tries to mimic 4 expert trajectories on a plane, each represented by a different color to indicate distinct modes (styles) of expert behavior (a small portion of the trajectories are labeled with one-hot behavioral categories).
>
> Fig. 1 (a) shows the expert trajectories, where the red and green trajectories are predominant, while the blue and purple trajectories only represent a small portion. Our Ess-InfoGAIL algorithm can successfully imitate distinguishable behavior styles, while GAIL and InfoGAIL can only imitate the more abundant red and green expert trajectories, and they cannot differentiate between trajectory styles. The corresponding descriptions have been added to Fig. 1. For more detailed information on expert demonstrations, please refer to Common Response A2.
>
> > Q2. The code is not available as of yet.
>
> A2. To enhance the readability of the code, we have organized it and uploaded it to an anonymous GitHub repository. In accordance with the rebuttal policy, which prohibits the inclusion of links to external pages in posted content, we have provided an anonymized link to the Area Chair (AC) in a separate comment. The code will be released to the public promptly after the paper's publication.
>
> > Q3. Table 1 is missing some measure of statistical significance.
>
> A3. Thank you for your valuable comment. Tables 1, 2, and 3, along with error bars (standard deviation) of the data, have been incorporated into the Appendix.
>
> > Q4. In line 263, two numbers are listed twice. Why is the improvement over both ablation variants the same for both cases?
>
> A4. Thank you for your valuable feedback. This is mainly due to the precision setting of the data. In the revised version, we have uniformly kept three decimal places throughout the manuscript to avoid unnecessary confusion.
>
> > Q5. Fig. 5 does not seem to agree with the discussion. Could both Fig. 5(a) and 5(b) have the same y-scale to make it easier to compare?
>
> A5. Thank you for your valuable advice. We have adjusted Fig. 5, and revised the discussion in Section 4.2 to eliminate any ambiguity.
>
> > Q6. A non-exhaustive list of errors/typos and stylistic improvements are provided.
>
> A6. Thank you very much for providing your valuable suggestions. We have carefully corrected the minor errors in the paper and incorporated the suggested changes.

---

> > ### Author Response · Authors · 2023-08-20
> > **Follow-up to Reviewer dvkr**
> >
> > Dear Reviewer, we would like to ask if your concerns have been addressed by our responses and supplementary materials. Thank you for your time.

---

### Official Review · Reviewer_18Ku · 2023-07-06

**Soundness:** 3 good
**Presentation:** 3 good
**Contribution:** 3 good
**Rating:** 5
**Confidence:** 4

**Summary:**

This paper focuses on semisupervised imitation learning with imbalanced data. Mainly, the approach extends InfoGAIL with a semisupervised learning architecture, inspired by ss-InfoGAN, where the latent variable is decomposed into a semisupervised part and an unsupervised part. The semisupervised part is defined as a learnable latent skill variable to deal with the imbalance in the data. Regularises Information Maximisation with a label prior is further used to improve the robustness against different models of behaviour and limited data. The proposed method is evaluated on a toy example as well as high-dimensional control tasks in Mujoco, like humanoid.

POST-REBUTTAL COMMENTS
The authors have addressed my comments satisfactorily. They presented further experimental results. Therefore I have increased my score from "borderline reject" to "borderline accept".

**Strengths:**

- Well-written paper with clear motivation and objectives. The authors have intention to share the code.
- Experiments are well designed to showcase the contribution. For example, the method is evaluated with different levels of imbalance in the data (Table 2) and with different numbers of behaviour modalities (Table 3).

**Weaknesses:**

- Some of the arguments in the paper are not clear to me. Looking at figure 1, do we want the agent to imitate the expert's different behaviour styles? or do we want agent to do the task? Why is learning different modes of behaviour important in this context? In addition, how is this achieved in the Mujoco environments? For example, how are different expert behaviours generated in the reacher environment? Why is classification relevant in this context?
- Experimental results are limited. Why are the results presented in terms of NMI and ENT only? How do we know the task performance? Expected cumulative reward should be presented or a video supplementary is needed to show that it is actually working. [56] is an image representation learning method, it doesn't make sense to use the same metrics in this paper where the goal is imitation learning. In addition, experiments regarding degree of data imbalance and learning more modalities are shown only in the reacher environment. We don't know whether the findings can be generalised to different environments.

**Questions:**

- What is the amount of data used in the experiments? Only the percentages of labelled data have been provided. However, still 0.1% can be too much, if the number of total samples is large, like 1M. Explain clearly how the experiments were designed. Also present the cumulative rewards to demonstrate the effective of the proposed method.

**Limitations:**

The authors briefly discussed the limitations. Potential negative impact does not apply.

---

> ### Author Rebuttal · Authors · 2023-08-09
>
> # Response to Reviewer 18Ku
>
> > Q1. Confused about Fig. 1. Do we want the agent to imitate the expert's different behaviour styles? Why is learning different modes of behaviour important in this context? How is this achieved in the Mujoco environments? Why is classification relevant in this context?
>
> A1. (1) Fig. 1 illustrates a simple 2D-Trajectory scenario, facilitating the visual comparison of algorithm performance. In the environment, an agent tries to mimic 4 expert trajectories on a plane, each represented by a different color to indicate distinct modes (styles) of expert behavior. Fig. 1 (a) shows the expert trajectories, where the red and green trajectories are predominant, while the blue and purple trajectories only represent a small portion. Our Ess-InfoGAIL algorithm can successfully imitate distinguishable behavior styles, while GAIL and InfoGAIL can only imitate the more abundant red and green expert trajectories, and they cannot differentiate between trajectory styles. The corresponding descriptions have been added to Fig. 1.
>
> (2) In the MuJoCo environment, taking Reacher as an example, the expert behavior styles can be the robot arm reaching different target points. In all experiments, we first pre-train K expert policies. Then the agents interact with the environment based on the trained policies to sample K expert trajectories with different styles as training data. In addition, to imitate real world motion capture data, the final expert data used for training is obtained by randomly sampling from the K expert trajectories. Please refer to Common Response A2 for more details.
>
> (3) Our method can learn disentangled multi-modal behaviors from raw imbalanced expert data, utilizing only limited labeled data to guide the classification. This enables the policy network to focus more on learning representations related to behavior categories. Therefore, semi-supervised behavior style classification plays a crucial role in our approach. Please refer to Common Response A1 for more details.
>
> > Q2. Experimental results are limited. Expected cumulative reward should be presented. NMI and ENT metrics are not necessary. Experiments regarding degree of data imbalance and learning more modalities are shown only in the Reacher environment.
>
> A2. (1) Thank you for your valuable suggestion. We have supplemented the average reward in both the Experiments section of the main paper and the Appendix. All the results demonstrate the advantages of the proposed method. Please refer to Common Response A3 for more details. Furthermore, as this work focuses on semi-supervised imitation learning, a majority of the training data are unlabeled. In this context, NMI and ENT remain crucial metrics to assess whether the agent has learned distinct behavior modes. Therefore, we retain them in the main paper.
>
> (2) Reacher serves as a relatively convenient environment for evaluating the impact of data imbalance and the number of behavior modes. However, in other MuJoCo environments like Humanoid, generating expert policies with diverse behaviors (e.g., walking, jumping, crouching, etc.) through manual reward engineering is challenging (since we need to pre-train the expert policy). Nevertheless, such behavior data can be easily obtained using motion capture technology in the real world. In our future work, we will consider directly utilizing motion capture data to learn richer human behavior modes, as demonstrated in reference [1].
>
> [1] Peng X B, Guo Y, Halper L, et al. Ase: Large-scale reusable adversarial skill embeddings for physically simulated characters[J]. ACM Transactions On Graphics (TOG), 2022, 41(4): 1-17.
>
> > Q3. Explain clearly how the experiments were designed in Amount of Labeled Data section. Also present the cumulative rewards to demonstrate the effective of the proposed method.
>
> A3. (1) Thank you for your valuable comment. If not specified otherwise, the default label ratio for the environments is as follows: 2D-Trajectory: 1%, Reacher: 0.5%, Pusher: 1%, Walker: 1%, Humanoid: 2%. For instance, in the Reacher environment, 0.1% of labeled data corresponds to 2 episodes (100 time steps) of each behavior mode. The labeled data used for training is very limited, covering only a few episodes in each environment of each behavior mode. More comprehensive details have been included in Section 4.2.
>
> (2) Thank you for your valuable suggestion. We have supplemented the average reward in both the Experiments section of the main paper and the Appendix. All the results demonstrate the advantages of the proposed method. Please refer to Common Response A3 for more details.
>
> We hope that the above response has addressed your concerns.

---

> > ### Comment · Reviewer_18Ku · 2023-08-16
> > **rebuttal acknowledgement**
> >
> > dear authors, this is just to let you know I read other reviews and responses. I do not require any further information at this stage, and need some time to process. if I need further information, I will touch base during the weekend. thanks.

---

> > > ### Author Response · Authors · 2023-08-16
> > > **Response to Reviewer 18Ku**
> > >
> > > Thank you for your acknowledgment and great efforts in helping us improve our work. We look forward to hearing more of your valuable insights and suggestions.

---

> > > ### Author Response · Authors · 2023-08-20
> > > **Follow-up to Reviewer 18Ku**
> > >
> > > Dear Reviewer, we wonder if you have any futher concerns regarding all responses. Thank you for your time.

---

> > > > ### Comment · Reviewer_18Ku · 2023-08-20
> > > > **re: Follow-up to Reviewer 18Ku**
> > > >
> > > > thanks. your responses address my comments. I don't have further comments.

---

### Official Review · Reviewer_fzCM · 2023-07-07

**Soundness:** 3 good
**Presentation:** 3 good
**Contribution:** 3 good
**Rating:** 7
**Confidence:** 4

**Summary:**

# Problem Statement

The paper addresses the problem of imitation learning in the context of real-world demonstrations that often present challenges such as multimodality, data imbalance, and expensive labeling processes.

# Main Contributions
The authors propose a novel semi-supervised imitation learning architecture, called Elastic semi-supervised InfoGAIL (Ess-InfoGAIL), that learns disentangled behavior representations from imbalanced demonstrations using limited labeled data. The method adapts the concept of semi-supervised generative adversarial networks to the imitation learning context, employs a learnable prior to align the generated and expert data distributions, and utilizes a regularized information maximization approach to further improve the semi-supervised learning performance. The authors demonstrate the efficiency of their method in challenging MuJoCo environments, showing that even in scenarios with highly imbalanced demonstrations, the policy can still reproduce the desired behavior modes.

# Methodology
The paper introduces a novel approach called Elastic semi-supervised InfoGAIL (Ess-InfoGAIL) for semi-supervised imitation learning from imbalanced demonstrations. The approach is based on three key improvements to InfoGAIL:

1. **Semi-supervised InfoGAIL**: The authors decompose the latent variable into a semi-supervised part and an unsupervised part. The semi-supervised part, denoted as '$c$', is a latent skill variable sampled from a categorical distribution, encoding the same information as the label '$y$'. The unsupervised part, denoted as '$ϵ$', is a latent shifting variable sampled from a continuous uniform distribution, allowing for style shifting within a given skill. The authors seek to maximize two mutual information terms $I(ϵ|s, a)$ and $I(c|s, a)$.

2. **Learnable Latent Skill Variable**: To align the state-action transitions produced by a policy $π_θ$ with the imbalanced expert demonstrations, the authors utilize a differentiable skill variable drawn from a Gumbel-Softmax distribution.

3. **Regularized Information Maximization (RIM) with an approximate label prior**: The authors leverage RIM with a label prior that approximates the learned latent skill distribution to make use of the intrinsic information in the unlabeled imbalanced data and improve the efficiency of semi-supervised learning.

The authors validate their method in a simple 2D trajectory environment for visualization and test it in four challenging MuJoCo environments.

# Experiments
The paper conducted experiments in a 2D trajectory environment and four MuJoCo environments (Reacher, Pusher, Hopper, and Humanoid) to validate the proposed method's ability to discover disentangled behavior representations from imbalanced demonstrations with limited labeled data. The experiments also analyzed the amount of labeled data required for the model to effectively encode the semantic meaning of the labels and the effect of varying the degree of data imbalance on the imitation of multimodal behaviors. The results demonstrated the efficiency of the proposed method, Ess-InfoGAIL, in learning multimodal behaviors compared to baseline methods such as GAIL, InfoGAIL, ACGAIL, and Elastic-InfoGAIL. The paper also conducted ablation experiments to verify the performance improvement of Gumbel Softmax and RIM techniques under the semi-supervised framework and imbalanced data.

**Strengths:**

# Originality and significance

The work innovatively introduces semi-supervised learning to online adversarial imitation learning of multimodal behaviors by extending InfoGAIL with mechanisms proposed in Ess-InfoGAN. The imitation learning from demonstrations that contain distributional multimodality is a topic of great interest, and the approach shows promising performance.

# Quality

The algorithm derivation and experiments are solid.

# Clarity

Overall the article is clear, although some details are not provided.

**Weaknesses:**

- Minor typo: The minimization should have $Q$ as a optimization variable in addition to $\pi$.

- The tasks in all experiments are not difficult due to the clear separation of the modes. As a reference, [1] shows that randomized search of the latent codes without encoder or posterior estimator can discern more subtly separated behavior modes in MuJoCo environments similar to those tested in this article (e.g. walker going forward and backward with various speeds).

- The evaluation metrics are limited. Only entropy and mutual information are reported, while for the environments and tasks appear in the article, there are commonly-acknowledged reward or score that can be used to quantify the quality of learned policy. Without those metrics reported, the quality of the policies is unclear to the readers.

- The implementation codes are not provided.

[1]: Vahabpour, A., Wang, T., Lu, Q., Pooladzandi, O., & Roychowdhury, V. (2022). Diverse Imitation Learning via Self-Organizing Generative Models. arXiv preprint arXiv:2205.03484.

**Questions:**

- In equation (9) and algorithm 1, which part corresponds to the learnable latent skill variable? What are the learnable parameters and what is their update in each iteration?
- What is $D_\text{body}$ in Figure 2?
- What is the label ratio in the data imbalance experiments described in section 4.3?
- How are the expert demonstrations generated?

**Limitations:**

The authors note that the limitations include the requirement of a small amount of labeled data for each category and the preset number of modal categories.

In addition, the limitation of low difficulty of tasks and the limited evaluation metrics are mentioned in the "Weakness" section.

---

> ### Author Rebuttal · Authors · 2023-08-09
>
> # Response to Reviewer fzCM
>
> > Q1. Minor typo: The minimization should have value function as a optimization variable in addition to $\pi$.
>
> A1. Thank you for your valuable suggestion. We have incorporated the optimization of the value function into Section 3.3 of the main paper.
>
> > Q2. The tasks in all experiments are not difficult due to the clear separation of the modes. Taking [1] as a reference.
>
> A2. (1) Learning distinguishable behavior modes from multi-modal data remains challenging, especially when the data size of each behavior mode is imbalanced.
>
> (2) The provided reference [1] is an intriguing work that utilizes a generator model without an encoder for behavior cloning, capable of distinguishing and imitating different behavior modes. However, fully unsupervised searching for latent variables has been proven to be difficult, as indicated in [2], which is evident from the exhaustive search for latent variables in the procedure of [1]. This is also the motivation behind our introduction of semi-supervised learning. In addition, our method can efficiently learn both discrete behavior modes and continuous behavior styles (e.g., walking speed) without the need for exhaustive search. Please refer to Appendix C for a simple visualized example. It is promising to combine and complement our work with [1]. Unfortunately, the code for [1] is not available yet.
>
> [1] Vahabpour A, Wang T, Lu Q, et al. Diverse Imitation Learning via Self-Organizing Generative Models[J]. arXiv preprint arXiv:2205.03484, 2022.
>
> [2] Locatello F, Bauer S, Lucic M, et al. Challenging common assumptions in the unsupervised learning of disentangled representations[C]//international conference on machine learning. PMLR, 2019: 4114-4124.
>
> > Q3. The evaluation metrics are limited. The average reward should also be provided.
>
> A3. Thank you for your valuable suggestion. We have supplemented the average reward in both the Experiments section of the main paper and the Appendix. All the results demonstrate the advantages of the proposed method. Please refer to Common Response A3 for more details.
>
> > Q4. The implementation codes are not provided.
>
> A4. To enhance the readability of the code, we have organized it and uploaded it to an anonymous GitHub repository. In accordance with the rebuttal policy, which prohibits the inclusion of links to external pages in posted content, we have provided an anonymized link to the Area Chair (AC) in a separate comment. The code will be released to the public promptly after the paper's publication.
>
> > Q5. In equation (9) and algorithm 1, which part corresponds to the learnable latent skill distribution? What are the learnable parameters and what is their update in each iteration?
>
> A5. (1) The latent skill variable $\mathbf{c}$, concatenated with the state, serves as an input to the policy, and the learnable latent skill distribution $p(\mathbf{c})$ is updated along with the policy. Specifically, the objective function of $p(\mathbf{c})$ aligns with the policy, which aims at minimizing the first two parts of $V_{Ess-InfoGAIL}$ in Equation 9.
>
> The revised objective function of Ess-InfoGAIL in Equation 9 is defined as follows:
>
> $\min_{\pi, Q^{\epsilon}, Q^{\mathbf{c}}, p(\mathbf{c})} \max_{D}(V_{InfoGAIL} - \lambda_{2}L_{IS} - \lambda_{3}L_{RIM})$
>
> And the update of $p(\mathbf{c})$ is as follows:
>
> $p_{i+1}(\mathbf{c})=\arg\min_{p(\mathbf{c})}(V_{InfoGAIL} - \lambda_{2}L_{IS})$
>
> We have revised Algorithm 1 and the description of Equation 9 to enhance readers' understanding.
>
> (2) In our model, the learnable parameters include $\theta, \beta, \phi, \psi, p(\mathbf{c})$, corresponding to the parameters of the policy, value function, discriminator, encoders and the latent skill distribution. Their update processes in each iteration are detailed in the revised Algorithm 1.
>
> > Q6. What is $D_\text{body}$ in Fig. 2?
>
> A6. $D_{\text{body}}$ serves as a shared backbone network for the discriminator $D_{\phi}$ , and the encoders $Q_{\psi}^{\epsilon}$ and $Q^{\mathbf{c}}_{\psi}$ , responsible for extracting features from state-action pairs. We have added explanations in Fig. 2 and included the architecture diagram of the backbone network in the Appendix.
>
> > Q7. What is the label ratio in the data imbalance experiments described in section 4.3?
>
> A7. In the data imbalance experiments of Section 4.3, the label ratio is set to 0.5%, where 0.1\% of the data corresponds to 2 episodes (100 time steps) of each behavior mode in the Reacher environment. We have included relevant descriptions in Section 4.3. Specifically, if not otherwise specified, the default label ratios are as follows: 2D-Trajectory: 1%, Reacher: 0.5%, Pusher: 1%, Walker: 1%, Humanoid: 2%. The corresponding descriptions have also been added to Section 4.2.
>
> > Q8. How are the expert demonstrations generated?
>
> A8. To collect imbalanced multi-modal demonstrations, we first train an expert policy for each mode. Subsequently, based on these expert policies, the agent interact with the environment to collect expert trajectories. Finally, we extract data from these trajectories in different proportions to create imbalanced multi-modal demonstrations. Please refer to Common Response A2 for more details.

---

> > ### Author Response · Authors · 2023-08-20
> > **Follow-up to Reviewer fzCM**
> >
> > Dear Reviewer, we would like to ask if your concerns around the related work and the evaluation metrics have been addressed in the author response. Thank you.

---

> > > ### Comment · Reviewer_fzCM · 2023-08-20
> > > **Thanks for the rebuttal**
> > >
> > > Thanks to the authors' rebuttal. The responses address my concerns and I will update my rating from 6 to 7.
> > >
> > > The Q1 was meant for the equation (3). Sorry for the confusion.
> > >
> > > Another minor issue is that I suggest not using math fonts for "Ess−InfoGAIL" and "InfoGAIL" in subscripts, as they are names instead of quantities. Upright roman or sans-serif fonts might be better options.

---

> > > > ### Author Response · Authors · 2023-08-20
> > > > **Follow-up to Reviewer fzCM**
> > > >
> > > > Thank you for your valuable comments. We have made the modifications based on your suggestions. Once again, thank you!

---

### Official Review · Reviewer_iRnn · 2023-07-08

**Soundness:** 3 good
**Presentation:** 2 fair
**Contribution:** 2 fair
**Rating:** 4
**Confidence:** 3

**Summary:**

The paper introduces a semi-supervised imitation learning architecture that addresses challenges associated with real-world demonstrations, such as multimodality, data imbalance, and expensive labeling processes. The proposed method utilizes three key components: adapting semi-supervised generative adversarial networks to the imitation learning context, employing a learnable prior to align generated and expert data distributions, and utilizing a regularized information maximization approach along with the learned prior to enhancing semi-supervised learning performance. Experimental results highlight the effectiveness of the proposed method in learning multimodal behaviors from imbalanced demonstrations, outperforming baseline methods.

**Strengths:**

* The idea of this paper is well-motivated and the investigated problem is practical.
* Empirical results show great improvement over baseline and other compared methods. The visualized results are good to demonstrate the efficiency

**Weaknesses:**

* The novel seems to be limited. This paper seems to be an extension of ss-InfoGAN [1] in the imitation learning fields. And the authors do not fully discuss the difference between ss-InfoGAN and the proposed method.
* The quality of the paper can be improved. There seem to be a lot of types in the paper. For example, the missing reference of the figure in line 208, 212, 216. Moreover, I also can not find the definition of $D_body$ in Figure 2, which makes it hard to understand the framework of the whole algorithm.

[1] Spurr, Adrian, Emre Aksan, and Otmar Hilliges. "Guiding infogan with semi-supervision." Joint European Conference on Machine Learning and Knowledge Discovery in Databases. Cham: Springer International Publishing, 2017.

**Questions:**

* Please provide more details about how to collect imbalanced multi-modal demonstrations used for training.
* I am also confused about the evaluation metrics. It seems that the authors only measure the distance between agent demonstrations and multi-modal expert demonstrations. However, the average reward, which is a basic evaluation metric should also be provided for comparison.

**Limitations:**

The authors have briefly discussed the limitations of the paper in the last section.

---

> ### Author Rebuttal · Authors · 2023-08-09
>
> # Response to Reviewer iRnn
>
> > Q1. The novel seems to be limited. More discussions with ss-InfoGAN needs to be added.
>
> A1. Indeed, we draw inspiration from ss-InfoGAN and extended it to the imitation learning framework, addressing key issues that still persist in the field of imitation learning, such as multi-modal behavior, data imbalance, and expensive labeling processes. The main differences from ss-InfoGAN are as follows: 1. We focus on imitation learning under sequential decision-making tasks, and 2. We introduce a learnable latent skill distribution and an improved RIM approach to tackle critical challenges commonly present in imitation tasks based on a substantial amount of imbalanced raw data. Relevant discussions have been added to section 3.2. Please refer to Common Response A1 for more details.
>
> > Q2. Some minor errors need to be corrected. The definition of $D_\text{body}$ in Fig. 2 is not clear.
>
> A2. (1) Thank you for your helpful comments. We have rectified these minor errors.
>
> (2) $D_{\text{body}}$ serves as a shared backbone network for the discriminator $D_{\phi}$ , and the encoders $Q_{\psi}^{\epsilon}$ and $Q^{\mathbf{c}}_{\psi}$ , responsible for extracting features from state-action pairs. We have added explanations in Fig. 2 and included the architecture diagram of the backbone network in the Appendix.
>
> > Q3. How to collect imbalanced multi-modal demonstrations used for training?
>
> A3. To collect imbalanced multi-modal data, we first train an expert policy for each mode. Subsequently, based on these expert policies, the agent interact with the environment to collect expert trajectories. Finally, we extract data from these trajectories in different proportions to create imbalanced multi-modal data. Please refer to Common Response A2 for more details.
>
> > Q4. The average reward should also be provided for comparison.
>
> A4. Thank you for your valuable suggestion. We have supplemented the average reward in both the Experiments section of the main paper and the Appendix. All the results demonstrate the advantages of the proposed method. Please refer to Common Response A3 for more details.

---

> > ### Author Response · Authors · 2023-08-20
> > **Follow-up to Reviewer iRnn**
> >
> > Dear Reviewer, we wonder if your concerns around the data collection and novelty of our method have been addressed in the author response. Thank you.

---

### Author Rebuttal · Authors · 2023-08-09

# Common Response

We are thankful to the reviewers for their valuable feedback. We first address the comments that are common to multiple reviewers and then response to the reviewers individually.

> Q1. The significance of this work.

A1. Our method draws inspiration from semi-supervised GANs and extends it to the multi-modal imitation task under imbalanced data, which plays a crucial role in enabling agents to effectively imitate real-world demonstrations. By utilizing the proposed Ess-InfoGAIL, high-quality multi-modal behaviors can be learned from a large amount of **raw expert demonstrations** without the need for:

* Labeling behavior modes (e.g., human walking, running, jumping, etc.) for each data point.
* Extracting independent segments of behavior modes.
* Establishing a balanced distribution of data among behavior categories.

All we require is a very limited amount of labeled data for learning guidance. These advantages are currently lacking in the majority of existing imitation learning algorithms.

> Q2. How to collect multi-modal demonstrations?

A2. In our experiments, we first pre-train K expert policies, each corresponding to K different goals (or K behavior modes). Subsequently, we use these K expert policies to sample K sets of expert demonstrations. From each set of expert demonstrations, we extract a small portion and label them with the one-hot behavioral categories, while the remaining expert demonstrations are randomly sampled and mixed to create imbalanced unlabeled expert data.

Moreover, in the real-world scenario, one can directly use the raw motion capture data (e.g., motion capture data of an animal over a day) without the need to train additional expert policies. The collection of multi-modal demonstrations has been clarified and added in Section 4 of the main paper.

> Q3. The average reward needs to be provided.

A3. We sincerely appreciate the valuable suggestions from the reviewers. We have incorporated relevant discussions of the average reward in section 4.1 of the main paper. The average reward of Reacher environment during the training process is added in Fig. 5. Due to space constraints, the average reward of other environments have been included in the Appendix. Furthermore, the normalized average reward quantification table, encompassing the outcomes of all algorithms, has been included in the Appendix, presented below. The results are also included in an additional single page PDF file. All the results demonstrate the advantages of the proposed method.

|  | 2D trajectory | Reacher | Pusher | Walker-2D | Humanoid |
|---|---|---|---|---|---|
| GAIL | 0.071$\pm$ 0.019 | 0.189$\pm$ 0.087 | 0.338$\pm$ 0.029 | 0.459$\pm$ 0.014 | 0.508$\pm$ 0.033 |
| InfoGAIL | 0.155$\pm$ 0.028 | 0.223$\pm$ 0.054 | 0.431$\pm$ 0.043 | 0.552$\pm$ 0.059 | 0.526$\pm$ 0.127 |
| ACGAIL | 0.540$\pm$ 0.061 | 0.616$\pm$ 0.131 | 0.790$\pm$ 0.042 | 0.703$\pm$ 0.040 | 0.614$\pm$ 0.040 |
| Elastic-InfoGAIL  | 0.189$\pm$ 0.059 | 0.261$\pm$ 0.079 | 0.483$\pm$ 0.042 | 0.616$\pm$ 0.039 | 0.544$\pm$ 0.111 |
| Ess-InfoGAIL$\backslash$GS  | 0.845$\pm$ 0.023 | 0.868$\pm$ 0.050 | 0.867$\pm$ 0.040 | 0.812$\pm$ 0.050 | 0.817$\pm$ 0.017 |
| Ess-InfoGAIL$\backslash$RIM  | 0.882$\pm$ 0.036 | 0.826$\pm$ 0.035 | 0.772$\pm$ 0.061 | 0.770$\pm$ 0.019 | 0.751$\pm$ 0.065 |
| Ess-InfoGAIL (Ours)  | **0.956$\pm$ 0.040** | **0.933$\pm$ 0.033** | **0.967$\pm$ 0.022** | **0.911$\pm$ 0.042** | **0.920$\pm$ 0.036** |

It is important to note that we make modifications to the original MuJoCo environment, and the computation of the task reward varies with different target behavior modes. We take the average value across all behavior modes as the final average task reward, and this task reward is only used as an evaluation metric and does not participate in the policy training process. Due to the issue of mode collapse and data imbalance, some methods (e.g., GAIL and InfoGAIL) may end up learning only one or two behavior modes, resulting in average task rewards across all modes that could be lower than those achieved by using a random policy.

> Q4. The implementation codes are not available.

A4. To enhance the readability of the code, we have organized it and uploaded it to an anonymous GitHub repository. However, in accordance with the rebuttal policy, which prohibits the inclusion of links to external pages in posted content, we have provided an anonymized link to the Area Chair (AC) in a separate comment. The code will be released to the public promptly after the paper's publication.

> Q5. Minor errors.

A5. We have corrected minor errors in the paper and have incorporated the suggested changes.

Please let us know if there are any remaining questions!

---

### Decision · Program_Chairs · 2023-09-21

**Decision:**

Accept (poster)

**Comment:**

This submission proposes an approach for semi-supervised imitation learning, as an extension to a popular work InfoGAIL. There were differing opinions among reviewers regarding the submission, especially in the initial version. On the positive side, all reviewers agreed that the idea of the submission is well-motivated and also the submission showed significant improvement over baseline methods. On the other hand, there were a few criticisms on the limited novelty of the method and the fact that the tasks in all experiments are not too difficult. The initial submission also lacked some key performance metrics (e.g. total reward-based comparison). However, the authors addressed most of the concerns fairly (including pointing out that the key performance metrics were already in the supp. material). Thus, all reviewers changed their opinions to be positive, and the AC also concurs with the overall feedback. Therefore, we think this submission is worth sharing with others in the community.